# Improving Fire Detection Accuracy through Enhanced Convolutional Neural Networks and Contour Techniques

**DOI:** 10.3390/s24165184

**Published:** 2024-08-11

**Authors:** Abror Shavkatovich Buriboev, Khoshim Rakhmanov, Temur Soqiyev, Andrew Jaeyong Choi

**Affiliations:** 1School of Computing, Department of AI-Software, Gachon University, Seongnam-si 13306, Republic of Korea; abror1989@gachon.ac.kr; 2Department of Infocommunication Engineering, Tashkent University of Information Technologies, Tashkent 100084, Uzbekistan; 3Department of Digital and Educational Technologies, Samarkand Branch of Tashkent University of Information Technologies, Samarkand 140100, Uzbekistan; hoshimrahmonov@gmail.com; 4Digital Technologies and Artificial Intelligence Research Institute, Tashkent 100125, Uzbekistan; temur.sakiev@gmail.com

**Keywords:** CNN model, fire detection, contour analysis, flame recognition

## Abstract

In this study, a novel method combining contour analysis with deep CNN is applied for fire detection. The method was made for fire detection using two main algorithms: one which detects the color properties of the fires, and another which analyzes the shape through contour detection. To overcome the disadvantages of previous methods, we generate a new labeled dataset, which consists of small fire instances and complex scenarios. We elaborated the dataset by selecting regions of interest (ROI) for enhanced fictional small fires and complex environment traits extracted through color characteristics and contour analysis, to better train our model regarding those more intricate features. Results of the experiment showed that our improved CNN model outperformed other networks. The accuracy, precision, recall and F1 score were 99.4%, 99.3%, 99.4% and 99.5%, respectively. The performance of our new approach is enhanced in all metrics compared to the previous CNN model with an accuracy of 99.4%. In addition, our approach beats many other state-of-the-art methods as well: Dilated CNNs (98.1% accuracy), Faster R-CNN (97.8% accuracy) and ResNet (94.3%). This result suggests that the approach can be beneficial for a variety of safety and security applications ranging from home, business to industrial and outdoor settings.

## 1. Introduction

Serious damage is caused by wildland fires. They result in air pollution, deforestation, desertification, economic losses, and firefighter and public fatalities. When the behavior of this significant risk can be predicted over time, the fight against it becomes even more successful. To comprehend and simulate the phenomena taking place during the spread of a fire, geometrical features such as position, rate of spread, length, surface, and volume are required. Camera-based frameworks have been created during the past ten years to serve as supplemental metrological instruments in fire experiments. Fire pixel detection is the first and most crucial stage in computer vision processing since it establishes the precision of the subsequent procedures. The primary challenges faced by detecting techniques in the visible range are caused by the color of fire and smoke. In fact, the hue can vary and be inhomogeneous based on the background and brightness; also, the fire zones may be superimposed by the smoke [1,2,3,4,5].

The literature has several potential fire detection algorithms [6,7,8,9,10,11,12,13,14]. CNNs have transformed the field of fire detection by greatly improving its accuracy, efficiency, and real-time capabilities. For instance, SmokeNet enhances smoke detection from satellite data for faster fire response [10]. Valikhujaev et al. developed an automated fire and smoke detection system using expansion filters and customized datasets [11]. Other notable approaches include Barmpoutis et al.’s fire detection system with wind sensors and dynamic scripts for early warnings [12], and various CNN-based methods by Li and Zhao [13] and Muhammad et al. [14] which improve situational awareness and detection flexibility.

Color rules in several color spaces, such as RGB, YUV, HSI, HSV, or a combination of other color spaces, are frequently used by algorithms in the visible spectrum [15,16,17,18,19,20,21]. Most of this research is applied in early fire detection scenarios, where real-time performance is critical. In this direction, one of the most pressing problems is the development and improvement of methods for implementing functions typical for most computer vision systems, such as preprocessing, detail extraction, segmentation and recognition. As part of solving this problem, the world scientific community has already obtained significant results of a theoretical and applied nature. At the same time, since the classical problem of computer vision for the general case, that is, for identifying arbitrary objects in random situations, has not yet been satisfactorily solved, it is necessary to improve existing, and develop new, more efficient, methods and algorithms.

Contours in images have rich information that has little dependence on color and brightness. When examining an object, a visual image is formed in the human mind. When perceiving, the eye tracks the contour line, which leads to the formation in the mind of an image with characteristic details. There is an opinion that during visual perception two images are formed: the contour and the inner part of the image. It is worth noting that the contour entirely determines the shape of the object and contains all the necessary information for classifying objects according to their shapes. Thus, this approach makes it possible not to consider the internal pixels of the object and to significantly reduce the area of processed areas due to the transition from analyzing a function with two parameters to a function with one parameter. It is for this reason that contour analysis methods can ensure the performance of a data processing system in real time. Even in those problems where it is impossible not to consider internal points, contour analysis methods complement the basic mathematical apparatus and, of course, are considered useful. The input for a typical image processing and analysis system is monochrome images of scenes containing objects of interest. To understand the content of the scene, it is necessary to recognize the objects located in the scene. The shape of an object is a binary image representing the size of the object.

Contour methods are aimed at identifying the boundary lines of objects that are clearly distinguished from the background. To identify the boundaries of objects, the criteria of the maximum modulus of the first derivative and the intersection of zero of the second derivative in the direction of the gradient of the brightness function are used. The resulting binary image is subjected to coding (for example, using the Freeman method) with the formulation of a vector of complex-valued numbers of the boundaries of objects and details of their internal structure. The main problem of contour analysis is the need to restore discontinuities in contours. Even so, contour methods have been used with great success for vectorization of multispectral images.

Advantages of contour methods in recognizing objects from satellite images:There is no need to form training and test samples for automatic recognition of individual classes.The contour method does not require the use of auxiliary approaches for localizing objects and constructing a feature space.Considering the size of multispectral satellite images (at least one hundred million pixels), the contour method provides minimal time for automatic vectorization.

Fire detection remains a critical area of research within the field of computer vision, particularly due to its implications for safety and emergency response. Traditional fire detection algorithms often rely heavily on color properties, which can lead to significant inaccuracies, especially in environments where fire-like colors are prevalent, but no actual fire is present. These limitations become apparent in scenarios such as dimly lit environments, reflections, and other visual artifacts that mimic fire characteristics.

Despite advancements, there remains a notable gap in the development of robust fire detection algorithms that can effectively differentiate between actual fires and fire-like conditions. Our study addresses this gap by integrating contour analysis with Convolutional Neural Networks (CNNs). This hybrid approach leverages the strengths of contour detection to capture the shape and movement of flames, thereby enhancing the accuracy of fire detection systems.

By creating a newly labeled dataset featuring small fire instances and complex scenarios, and combining it with sophisticated image processing techniques, our proposed method significantly improves detection accuracy. The results demonstrate superior performance compared to existing state-of-the-art methods, highlighting the potential for our approach to enhance fire detection across various applications.

### Problem Formulation and Motivation

Fire detection systems are essential to maintain safety and reduce damage in different areas. The availability of computer vision and machine learning has greatly improved fire detection skills, allowing systems to detect fire hazards faster and more accurately. Using a huge dataset, we first developed the CNN model. Some models for fire detection show good accuracy [22,23,24,25,26]. Even with encouraging results, the models had serious limitations, especially when it came to distinguishing real fire from fire-like images in certain situations, i.e., a dimly lit night system or a fire-like light mistaken for a real fire, as seen in Figure 1.

We carefully reviewed and improved our dataset to get beyond these challenges, and we also used color and contour analysis approaches to improve the CNN model. Reducing false positives and improving overall detection accuracy meant enhancing the model’s ability to distinguish between images with and without fire.

In order to greatly increase the size of the dataset, the first steps in the research process were to gather publicly accessible images and enhanced them using a variety of computer vision methods. Although this method produced better results, it was still unable to handle intricate situations involving things that resemble fire.

A novel approach to dataset creation and CNN model improvement was suggested by us in recognition of these flaws. To achieve this, a larger collection of images was gathered, with a fair distribution of both fire and non-fire photos, and sophisticated image processing methods including contour and color analysis were used. By combining those techniques, the version was purported to come to be extra correct at spotting fire in plenty of hard-to-locate situations.

This paper proposes an up-to-date CNN version architecture, a redesigned dataset production procedure, and the incorporation of color and contour evaluation. We include an intensive explanation of the education technique, evaluation criteria, and dataset augmentation techniques that had been employed to gauge the version’s effectiveness. The model’s accuracy and robustness have considerably advanced, in step with our findings, making it a more sincere device for fire detection in lots of real-world applications.

## 2. Related Works

### 2.1. AI-Based Approaches for Fire Detection

Pan and so on [27,28,29,30,31] presented a computationally efficient forest fire detection method using deep CNN and faster R-CNN optimized by Fourier analysis. The proposed method improves the performance of complex combinations by effectively detecting forest fires in surveillance photographs while preserving important distinguishing features. A very large convolutional community for wildfire smoke detection was created via Li et al. [32]. Their technique works nicely for identifying smoke styles in complex mental and environmental contexts, in consideration of rapid remedial and mitigation measures. Real-time fireplace facts received from a fireplace detection gadget using Kim Lee’s deep gaining knowledge of model [33] highlighted the effectiveness of CNNs in deciphering video streams for detection. Their method provides the rapid detection necessary for rapid response and containment of a fire. An automated fire detection and alarm system, improved by an upgraded YOLOv4 model, was proposed by Mukhiddinov et al. [34,35] with the express purpose of assisting the blind and visually impaired by providing timely warnings during fire occurrences. This technology provides quick notifications for prompt evacuation and assistance, which increases accessibility and protection. The forest fire detection and notification system was proposed by Avazov et al. [36,37]. To reduce the number of wildfires, researchers have developed new technology that uses artificial intelligence and Internet of Things devices and sensors. As a result, they believe that the proposed strategy can be used to effectively stop the death toll and rapidly worsening global climate problem. Installing the system in a forest will allow it to detect smoke, allowing the artificial intelligence model to determine the exact location of the fire and alert the fire brigade to extinguish the fire before it becomes a lasting issue. 

Due to its strong model structure and efficient feature extraction techniques, the CNN has performed better than other approaches in the fields of image and video classification. Thus, deep learning methods outperform classical computer vision methods in terms of performance. Our proposed approach uses a model to classify fires in images and videos. The rise in false fire alarms is due to misclassification of photos or videos due to differences in brightness, shadows and perspective distortions. Using a model that combines color characteristic algorithms and edge analysis algorithms with the CNN to learn and extract reliable aspects of a frame, we were able to recognize photographs containing fire.

### 2.2. Techniques in Image Processing for Identifying Fire

Image processing techniques have advanced beyond AI-driven methods to become important in the detection of smoke and fire in video sequences. By employing various algorithms and techniques, these methods seek to increase the accuracy and consistency of detection in a variety of environmental conditions. Here, we examine several studies that have advanced this field.

An automated computer vision system for fire detection in films was presented by Dimitropoulos et al. [38]. The system uses background subtraction and nonparametric color analysis to identify possible fire zones. It then assesses spatiotemporal features such as color likelihood, flicker, and energy. Subsequently, the bag of systems approach and linear dynamic systems are used for dynamic texture analysis. Sporoceptal coherence energy is computed with historical data to improve robustness. Gagliardi et al. [39] created AdViSED, a video-based smoke detection system that makes use of M of N decisions, color analysis, picture segmentation, blob labeling, geometrical feature analysis, and a Kalman estimator. By detecting smoke in a matter of seconds, this approach outperforms fire-alarm requirements such as EN50155. Based on Toulouseet’s study [40] measuring the geometrical properties of wildland fires, this research compares state-of-the-art image processing-based fire color identification rules and techniques. The test runs on approximately 200 million pixels with fire and 700 million pixels without fire, taken from 500 wildlife photographs under various shooting conditions. Non-burning pixels are classified based on the average intensity of the corresponding image, and burning pixels are classified based on the color of the fire and the presence of smoke. 

Wavelet analysis has been commonly used in video-based fire detection methods to determine whether a given pixel is in a fire zone, and Fast Fourier Transform has been used to characterize the contours of a fire zone by Zhang et al. [41]. They experiment with these two methods on photographs of wildfires and create a new method that combines FFT and wavelet analysis. First, if the outline of the fire is found, then it is displayed using FFT. Finally, temporal wavelet analysis is used to examine the FFT descriptors of each frame of the video clip. In addition to detecting fire frames more accurately than the wavelet method, this strategy avoids the need to set an edge threshold in the FFT method. Several wildfire videos are used to test the unique approach, and experimental results show promising results.

Celik et al. [42] investigated the use of a real-time adaptive background subtraction method to help segment fire candidate pixels from the background, which is the paper’s primary contribution. The fire detection system is created by combining the two methods, and it is used to identify fire in successive frames of video sequences.

Prema and so on [43,44,45] segmented images according to the color of the so-called flame candidate region in the YcbCr color space. 

The fact that these methods rely on data that is readily apparent when identifying fires in image frames is one of their problems. Factors that determine the occurrence of a fire include its color, speed of movement, surroundings, size, and edges. These techniques are hampered by foggy skies, poor image and video quality, and inclement weather. As a result, it is critical to improve these methods using the latest supporting strategies. In addition, our suggested approach—which is covered in Section 3—improves fire detection by including contour analysis and color characteristics methods. This technique enhances current image processing methods by adding contour-based characteristics to increase the precision and dependability of detection in surveillance and monitoring applications.

## 3. Proposed Method

In this work, to quickly and accurately detect fires, we integrated the image processing and deep learning advantages into one model, i.e., color and contour analysis ability of image processing and feature extraction, and the learning ability of deep learning models. This integration helped to increase accuracy in fire detection and minimize the false positives of previous CNN models. The Block scheme of the proposed hybrid model is presented in Figure 2.

The suggested framework consists of the following submodules:

Input image: this part receives fire and fire-like images at any resolution, including small-size. 

Color characteristics analyzer: this block checks and analyzes potential fire properties of an image and selects regions of interest if it detects fire or fire-like zones. For isolation of fire and fire-like zones, strategies like filtering and thresholding algorithms can be used.

Contour Analysis: this block works simultaneously with the color characteristics algorithm block and attempts to detect fire and fire-like zones from the input image using contours. It uses filters based on area, aspect ratio and convexity.

Resize image: this block receives potential fire images which are selected only ROI of potential fires and, regardless of their size, it can increase or decrease the image size. Moreover, the resizing process helps to accelerate the CNN model’s performance by focusing on the ROI of fires and improves its accuracy.

CNN model: The developed CNN is trained using a dataset which is developed as described in the below section. Using attributes taken from the scaled images, the CNN model learns to differentiate between areas that are on fire and those that are not.

Output: The model’s output is a prediction of the location of identified fire ROIs and whether there are any regions of fire in the input image.

The proposed hybrid model integrates the strength of image processing methods for quickly detecting and selecting potential fire regions with the CNN model’s advantages like feature extraction, learning abilities and high accuracy. 

Our development and training workflow involved the following steps:Data Preprocessing: Using OpenCV 4.8.1 for image resizing, normalization, and conversion to HSV color space. Contour detection and filtering were also performed at this stage.Model Construction: Leveraging Keras 2.13.1 to define the CNN architecture, including convolutional, pooling, and dense layers, along with activation functions and dropout for regularization.Model Training: Utilizing TensorFlow 2.15 and Keras to compile the model, specify the optimizer (Adam) and for loss function (binary cross-entropy) and metrics (accuracy). The model was trained using a GPU for accelerated computation.

We built and tested the suggested setup using the Anaconda 2020 Python distribution on a PC featuring two Nvidia GeForce 1080Ti GPUs (Nvidia, Santa Clara, CA, USA), 32 GB of RAM and a 3.20 GHz CPU

### 3.1. Fire Dataset

One main aspect of CNN model’s adequacy is dataset. If the model is trained with adequate data, the performance of the model will be satisfactory. Therefore, there are two requirements for dataset from the neural network, that is availability of sufficient amount of data and reliability of each data. To provide the CNN model with dataset as mentioned above we developed a more extensive and meticulous dataset. The developed dataset helps to overcome the limitations of previous models, which had confusion with fire-like images in nighttime environments, blurred lamps and bulbs. The processes and steps of dataset development are as follows: 

Firstly, we gathered two types of images from publicly open access datasets. One is fire images in different conditions and the second is fire-like images. The first image collection consists of 10,200 images, and the fire-like image collection has 10,120 images. The images have diverse sizes, hues and shapes of fire. Moreover, fire like image collection is included with hazy lightbulbs and lights. These types of images help to reduce the false positives of CNN model.

Secondly, to increase the number of images, we used data augmentation techniques using image processing techniques. It is designed to differentiate the images. Here, we rotated each image 17 times by 10° from 0° to 180°. In our previous work, we rotated the images at 0°–360°. This resulted in the creation of unrealistic fire images, i.e., an inverted image, which led to an increase in the CNN model’s false positive metric. The range of 0°–180° is chosen to prevent unrealistic fire image generation in the dataset; this range of rotation provides natural fire behavior, which typically involves rising and spreading sideway during the influence of wind. 

After 17 rotating processes, the dataset contained 173,400 fire images and 172,040 fire like images. The total dataset contained 345,440 images and was divided into a training set, testing set, and validation set; respectively, 70%, 20%, and 10%. 

### 3.2. Algorithm for Detecting Fire by Color Characteristics

Fire detection by color was one of the first recognition methods and is still used in most devices. The algorithm for recognizing fire by color characteristics includes the following steps:

Step 1. Convert the original RGB image into the HSV color space. The HSV color scheme stands for Hue, Saturation, and Value. The formula for converting the RGB color scheme to HSV is as following equation:


*
**Hϵ[0, 360], S,Vϵ[0, 1], max = max(R,G,B), min = min(R,G,B), V = max, S = (max − min)/max,**
*

$$H=60\;\times \;\left\{\begin{array}{c} 0+\frac{G-B}{max-min}\;\;\;\;\;\;\;\;\;if\;\;\;\;\;max=R,\\ 2+\frac{B-G}{max-min}\;\;\;\;\;\;\;\;\;if\;\;\;\;\;max=G,\\ 4+\frac{R-G}{max-min}\;\;\;\;\;\;\;\;\;if\;\;\;\;\;\;max=B, \end{array}\right.$$

H = H + 360  if  H < 0


Advantages of converting to HSV:Improved Fire Segmentation: Fire typically has a distinct hue range (red, orange, yellow). By converting to HSV, we can effectively isolate the hue component, allowing for more accurate segmentation of fire regions based on color.Robustness to Lighting Conditions: The separation of hue from value means that the detection algorithm becomes more robust to changes in illumination. This is crucial in real-world scenarios where lighting conditions can vary widely.Enhanced Contour Analysis: The conversion aids in contour analysis by providing a clearer distinction between fire and non-fire regions. The hue component helps in identifying the fire’s boundaries more accurately, which is essential for the subsequent contour-based processing.

By converting to the HSV color space, our method gains improved reliability and accuracy in detecting fires across diverse and challenging environments.

Step 2. Create a mask filter in the HSV color space. The ranges of the upper boundary HSV *upperBound* = [145, 255, 255] and the lower boundary *lowerBound* = [0, 0, 200] pixels are determined.

Step 3. After obtaining the binary image in the second step, the boundary line detection procedure is performed based on the Canny edge detector.

Step 4. To filter small noise pixels, a morphological operation procedure called “opening” and “closing” is performed. The “opening” operation performs a procedure for clearing small noise pixels that are outside the object’s outline, and the “closing” procedure, on the contrary, clears noise pixels inside the object’s outline.

Step 5. Contour coding is performed using the Freeman method [46], where the boundaries of edge points are formulated as chains of vectors of complex numbers that are invariant to displacement, rotation and scale.

Step 6. All found contours are placed in rectangular frames and marked with numbers.

Step 7. The conditions are checked—the flame detector must register flame occupying regions.

### 3.3. Algorithm for Recognizing Fire Based on Object Contours

After the video image is received from CCTV cameras, it is transmitted to the image processing and analysis unit. At the preliminary stage of recognition, the procedure for detecting fire is performed using the detection algorithm based on color characteristics given above.

Gonzales et al. [47] in his monograph “Digital image processing” indicates that for the correct identification of smoke and fire in images, one feature, which is color, is not enough, because this leads to an increase in false positives. This is because there are many objects of a similar color to the flame (for example, yellow leaves on trees or an orange sun at sunset).

A significant difference in these cases is the type of movement of dynamic objects. In the interval between two adjacent frames, the appearance of the fire can change dramatically, being at a certain point only at the exact time. Based on this, to correctly detect a flame, you need to use properties based on moving various frames *D*(*x*, *y*) in accordance with the factors that determine the color of the fire. Based on the arithmetic mean modules of the difference between scenes of a video sequence of the same point, moving objects are determined
$$D(x,y)=\frac{\sum_{t=0}^{n}\left|J\left(I_{t}\left(x,y\right)\right)-J(I_{t-1}(x,y))\right|}{n-1}$$
where *J* is a function that, for given values (R, G, B) returns a value equal to (R + G + B)/3.

When detecting a flame, the main problems are the reflections of nearby objects with a source of flame present. The destruction procedure can remove a large portion of such reflections. In this case, for each flame point, eight points located next to it are tracked, and when suddenly no more than half of them turn out to be flame points, then the tracked point is not considered as a fire pixel [47].

The structure of the fire has a certain color. If the center of the fire is brightly transparent, then moving towards the boundaries of the hearth, the color changes from blue to red, orange, and yellow. In monochrome photographs, the core of the image is significantly brighter than the periphery. You should also keep in mind that the sphere of flame has several separate bright parts. The sphere of flame in a certain part of the image can be considered as a large contrast taken with adjacent areas, and wedged into the spatial structure of color, starting with white in the middle and ending with red at the border, as shown in Figure 3.

Detection of suitable flame spheres is, in most cases, implemented in monochrome images using individual areas with high density. Separately detected areas are then increased towards the spectrum gradient by adding adjacent points where colors have a high probability in accordance with the selected color palette. The tolerance intensity of the interior of the sphere is considered by a mixture of Gaussian distributions in a multi-color palette (e.g., HSV). The developed recognition algorithm by the contour method is supplemented by checking the conditions for the previous algorithm to reduce fire detection errors.

Step 1. To separate the subject from the background in the presence of noise, it is necessary to perform a raster binarization procedure, the result of which will be a division of raster pixels into two classes: object and background. Let R = {r_m,n_} m = $\bar{1,M}$, n = $\bar{1,N}$ − image containing a separate dynamic object, then the rule for binarizing this image will look like:$${\hat{r}}_{m,n}=\left\{\begin{array}{c} 0\;if\;r_{m,n}\leq \beta \\ 255\;if\;r_{m,n}>\beta \; \end{array}\right\}$$
where $\hat{R}=\{{\hat{r}}_{m,n}\}$—binarized image, β—threshold value, which is selected based on the histogram of pixel brightness distribution *H-{h_k_*}, and *k* = $\bar{0,255}$ current raster.

Step 2. The contour of the boundary lines is formed according to the Beetle algorithm [48]. The method consists of sequentially drawing the boundary between the object and background. A tracking dot shaped like a “beetle” crawls across images until it reaches the dark area (object). Then the “bug” turns to the left and moves along a curve until it reaches the boundaries of the object, then turns to the right and repeats the process, until it reaches the vicinity of the starting point as shown in Figure 4.

Thus, the output of the “Beetle” algorithm will be the formed contour of the selected object, which is a vector of complex numbers $H=\{\gamma_{k}\}$, $k=\bar{0,K-1}$.

Step 3. When detecting the outlines of dynamic objects, the following condition is checked: whether the boundary of the external outline with the inner part of the area, in which most of the pixels have very high intensity (completely white areas). The shape of the fire area usually changes constantly, and in doing so carries out stochastic movements depending on external factors such as type of burning material and air flows covering the fire source.

Figure 5 demonstrates the effectiveness of the contour-based detection algorithm. The detected fire regions are outlined, showing two distinct rings indicating the fire’s core and its periphery. The inner ring corresponds to the brightest, most intense area of the fire, while the outer ring encompasses the broader fire region, including lower intensity flames. The algorithm effectively differentiates the fire from surrounding objects, even those with similar colors, by utilizing dynamic movement and contour analysis. This approach reduces false positives and enhances detection accuracy, as shown by the clear and distinct fire outlines in the result image.

### 3.4. Architecture of CNN Model for Fire Detection

The most important component of our proposed hybrid model is CNN layers. The aim of using this component is to integrate the advantages of CNN models, which are high speed, efficient and allow for accurate detection of fires between real fires and non-fires. Figure 6 shows the CNN model used in our fire detection system. It consists of several interconnected layers, each of which performs a specific function in processing and analyzing image data. The architecture allows the model to learn and recognize patterns suggestive of fire by gradually extracting abstract features from input photos.

Here, we provide a detailed description of the model architecture and training parameters.
Input Layer: The ROI images that were shrunk by the color attributes and contour analysis methods that came before are accepted by this layer. These input images typically have dimensions of 256 by 256 pixels, which was selected to strike a compromise between computing efficiency and maintaining enough information for precise categorization.Convolutional Layers:
First Convolutional Layer: Applies 32 filters with a 3 × 3 kernel size, and then an activation function called ReLU (Rectified Linear Unit). The primary objective of this layer is to extract fundamental elements from the input images, like edges and simple textures. The size of the feature map in output is 126 × 126 × 32.Second Convolutional Layer: Applies 64 filters, each having a 3 × 3 kernel, and then the ReLU activation function. By expanding on the results of the previous layer, this layer can capture more intricate details. Because of max pooling, the feature map’s size is lowered to 62 × 62 × 64.Third Convolutional Layer: Applies an ReLU activation function after 128 filters with a 3 × 3 kernel size. This layer further abstracts the visual data by identifying high-level properties like certain patterns associated with fire. After pooling, the feature map measures 30 × 30 × 128.
Pooling Layers:
A max pooling layer with a pool size of 2 × 2 and a stride of 2 follows each convolutional layer. The feature maps are down-sampled by these layers, maintaining the most important features while decreasing the feature maps’ spatial dimensions and computing burden. Preventing overfitting and enhancing the model’s capacity for generalization are dependent on this stage.
Fully Connected Layers:
First Fully Connected Layer: Initial Completely Networked Layer: consists of 256 neurons that have been activated by ReLU. It allows the model to perform intricate feature interactions by converting the 3D feature maps into a 1D feature vector. To combine the extracted features into a more abstract representation, this layer is essential.Second Fully Connected Layer: Consists of 128 neurons that have been activated by ReLU, which helps to improve the feature abstraction and get the data ready for the last classification layer.Output Layer: provides a probability score for binary classification (fire or non-fire) using a SoftMax activation function. The likelihood of each class is indicated by a vector of two probabilities that this layer outputs. The model’s predictions are comprehensible since the SoftMax function makes sure that the probability adds up to 1.


Training Parameters of proposed CNN model:Loss Function: Binary Cross-Entropy, suitable for binary classification tasks.Optimizer: Adam optimizer with an initial learning rate of 0.001, known for its efficiency and adaptability in deep learning tasks.Batch Size: 32, balancing computational load and convergence speed.Number of Epochs: 50, allowing sufficient time for the model to learn while avoiding overfitting.Early Stopping: Implemented to monitor the validation loss, with a patience parameter set to 10 epochs, stopping training when the validation loss does not improve.

These parameters were determined through extensive experimentation to strike an optimal balance between computational efficiency and detection accuracy. The convolutional layers are designed to extract intricate features related to fire, while the pooling layers help in reducing dimensionality, thus improving the robustness of the model. The fully connected layers, along with dropout, ensure the model captures relevant features without becoming overly complex.

Our hybrid technique combines the CNN model with preprocessed ROIs from the color characteristics and contour analysis algorithms, greatly increasing efficiency and accuracy in fire detection. The shortcomings of conventional techniques are addressed, and the overall dependability of the fire detection system is increased, thanks to the CNN’s capacity to learn intricate patterns and features that enable it to distinguish between fire and fire-like things. This hybrid system offers a reliable solution for practical uses, marking a substantial improvement in fire detection technology.

## 4. Experimental Results and Discussions

This section presents the experimental results of our hybrid fire detection system, which combines traditional image processing techniques with a CNN to improve accuracy. Even though our previous model had an impressive 97.7% accuracy, we had trouble identifying small fires. We made an improvement to overcome this limitation and further improve the performance of our model. Adding tiny photographs to our dataset was one of the main tactics that helped our machine identify and learn detailed characteristics associated with small fires.

In this paper, we included small sized images which contain only fire, i.e., each image has only fire and does not include background information. By integrating the dataset with these kinds of images, our CNN model is provided more exact data which impacted the learning of details and allowed it to work more accurately during the detection of fires. 

The efficiency of our model was evaluated using a great deal of testing and verification. In the experiments, we assessed the model’s performance by standard metrics like Precision, Recall, F1 and accuracy in different datasets with and without small size fire images.

The performance and results validate the significant improvement of our proposed CNN model, especially in small size fire detection as shown in Figure 7. We observed a notable increase in accuracy and precision, indicating the successful integration of the large-scale feature map and feature concatenation techniques.

The proposed flame detection method was tested on a sample of twelve videos with various scenes, including videos filmed during the day and night, as well as indoors and outdoors. Experimental results on the operation between the proposed CNN model and image processing algorithms are displayed in Table 1. Frames of laboratory experiments are shown, demonstrating the operating conditions of the model. The fourth and fifth columns of Table 1 show errors of the first and second types. Type 1 error (FRR), displays omission errors, meaning that the system does not detect a flame in a video frame when there is a fire in it. Type 2 error (FAR) is the number of frames that cause a false alarm, that is, the system detects a fire in a video frame when there is no fire. The last column of the table presents the performance of the CNN model, and it shows much more accurate results than color and contour techniques.

The efficiency of flame detection in video is defined as the ratio between the number of correct alarms and the number of all video frames:$$R=\left(1-\frac{S_{lost}}{S_{frame}}\right)\times 100\%$$
where *S_frame_* is the total number of frames in the video, *S_lost_* is the sum of errors of the first and second kind and R is the result of detection in (%).

Contour analysis plays a crucial role in our fire detection methodology by providing an additional layer of feature extraction that complements the capabilities of the CNN model. Traditional fire detection methods relying solely on color features often encounter challenges in differentiating between fire and fire-like objects, especially in complex environments. Contour analysis addresses this limitation by focusing on the shape and boundary information of the objects within the images. The integration of contour analysis allows our model to accurately identify the edges and shapes of flames, which are distinctive compared to other objects with similar color characteristics. This process involves detecting the boundary lines of objects and using the maximum modulus of the first derivative and the intersection of zero of the second derivative in the direction of the gradient of the brightness function. The resulting binary image, coded with contour information, significantly enhances the CNN’s ability to distinguish true fire instances from false positives.

In our experiments, we observed that the combined use of color and contour analysis led to substantial improvements in detection metrics. The contour analysis algorithm effectively reduced the false positive rate and false negative rate by providing additional discriminative power to the CNN model. For instance, in scenarios involving fire-like objects such as lamps at night or individuals in yellow-red clothing, contour analysis helped the model accurately classify these instances, reducing misclassifications.

The table presents a comprehensive evaluation of fire detection performance using two different approaches: a traditional color and contour analysis algorithm and a CNN. The data spans a variety of fire scenarios, illustrating the effectiveness and limitations of each approach. For instance, in the “Fire in the kitchen” scenario, the CNN achieved a perfect accuracy rate of 100%, significantly outperforming the color and contour analysis method, which recorded a detection accuracy of 95.2%. This trend continues across other scenarios, highlighting CNN’s robust performance. In “Fire in the supermarket”, the CNN again demonstrated superior accuracy, reaching 99.8% compared to the traditional method’s 97.8%. Similarly, for “Fire in the forest”, CNN achieved an impressive 99.9% accuracy, surpassing the Color and Contour Analysis algorithm’s 97.4%.

The CNN method’s reliability extends to more challenging scenarios as well. For instance, in “Day lamp and fire”, where distinguishing between fire and non-fire objects could be particularly complex, the CNN maintained a high accuracy of 99.8%, while the traditional approach lagged with a 93.6% accuracy. This superior performance is also evident in “Fire and a man in yellow-red clothes”, where the CNN achieved a 99.9% accuracy compared to the traditional method’s 94.9%.

In scenarios involving dynamic or variable environments, such as “Fire on the highway”, the CNN reached an accuracy of 99.8%, while the color and contour analysis algorithm struggled, with a 91.4% accuracy. Even in challenging conditions like “Lamp at night” and “explosion”, where distinguishing fire from other light sources is difficult, the CNN method excelled with accuracies of 99.8% and 100%, respectively, compared to the color and contour analysis algorithm’s 97.7% and 97.1%. Overall, the data reveals that the CNN approach significantly enhances fire detection capabilities across diverse and complex scenarios. It not only reduces false rejection rates, but also minimizes false acceptance, making it a more reliable and accurate tool compared to the traditional color and contour analysis method. This suggests that the CNN method is highly effective in improving fire detection performance, making it a valuable advancement in fire detection technology.

### Performance Metrics and Comparative Analysis

The performance of our method is compared against several well-known deep learning models: Dilated CNNs, AlexNet, Faster R-CNN, ResNet and VGG16. The evaluation metrics used include Precision (P), Recall (R), F1 Score (FM) and the Average Accuracy. The comparative results are summarized in Table 2.

The precision metric quantifies the percentage of accurately detected fire incidents among all the incidents categorized as fire. Reducing false positives is critical since they can cause needless alerts.

Recall quantifies the percentage of real fire incidents that the model accurately classified. A high recall rate is necessary to guarantee that all fires are found, reducing the possibility of fires going undiscovered.

The F1 score provides a single metric that balances the trade-off between precision and recall, as it is the harmonic mean of these two metrics. It is especially helpful in situations where recall and precision are crucial.

The Average Accuracy statistic offers a broad assessment of the model’s efficacy by measuring its performance across all classes.

Figure 8 illustrates the performance comparison of various fire detection algorithms across four metrics: Precision (P), Recall (R), F-Measure (FM), and Average percentage. Each algorithm is represented along the x-axis, with their respective performance metrics plotted as lines with distinct markers. By visualizing the data in this manner, it is easier to compare the performance of each algorithm across the different metrics, highlighting the superiority of the proposed CNN, which consistently scores highest across all parameters. The Dilated CNNs and our previous CNN also demonstrate strong performance, whereas AlexNet shows the lowest performance across most metrics. This graphical representation emphasizes the relative strengths and weaknesses of each model, aiding in clearer comparative analysis.

As seen in Figure 8, the addition of contour analysis to an improved dataset curation technique has greatly increased our model’s capacity to detect fire incidents with few false positives and negatives. To avoid false alarms, high precision (99.3%) ensures that the cases labeled as fire are nearly real fires. A high recall rate of 99.4% ensures that almost all real fire incidents are found, which is crucial for both safety and prompt action. Our method’s F1-Score of 99.5 percent suggests a well-balanced performance, combining the advantages of high recall and high precision. In real-world situations, where both missing a fire incident (low recall) and setting off a false alarm (low precision) can have serious repercussions, this balanced approach is essential. Our approach outperforms earlier algorithms, including our previous CNN model. Despite the prior model performing well, the additional improvements—such as improved dataset curation and sophisticated CNN architecture—made it much better. The resilience and reliability of our technique is demonstrated by the significant improvement in average accuracy (99.4%) over existing high-performing models such as ResNet (94.3%) and Dilated CNNs (98.1%). Although Dilated CNNs and VGG16 are also good algorithms, our technique achieves better precision and recall. For instance, Dilated CNNs have a modest decrease in recall but a high precision. In contrast, AlexNet performs much worse across all parameters, demonstrating its inability in intricate fire detection tasks in comparison to more sophisticated architectures such as ours. With an F1 score of 99.5%, a recall of 99.4% and a precision of 99.3%, our hybrid approach comes in top on all metrics. 99.4% accuracy is the average. The exceptional results may be ascribed to the proficient fusion of conventional image processing methods for the first ROI extraction and a thoughtfully constructed CNN for the ultimate classification.

By contrasting our model’s output with earlier research in the subject, we were able to further validate it. Table 3 provides a summary of the comparative outcomes.

Table 3 compares the performance of various fire detection algorithms based on precision (P), recall (R), F-measure (FM) and average accuracy. The proposed method achieves the highest scores across all metrics, with a precision of 99.3%, recall of 99.4%, F-measure of 99.5% and an average accuracy of 99.4%, significantly outperforming all previous approaches. Abdusalomov et al. [1] also performs well with high scores (P: 98.3%, R: 99.2%, FM: 99.5%, Average: 98.9%), indicating superior detection and minimal false positives. Valikhujaev et al. [11], Panagiotis et al. [49], and Renjie Xu et al. [53] show strong, balanced performances, though slightly lower than the proposed method. Redmon et al. [50] demonstrate reliable results but with room for improvement. Fei Shi et al. [51] shows the lowest precision and average accuracy, suggesting higher false positives and less consistent performance. Chengzhi Cao et al. [52] and Byoungjun et al. [33] provide good results but are outperformed by the proposed method and others. The line graph as shown in Figure 9 underscores the proposed method’s clear advantage, visually highlighting its superior performance in precision, recall, F-measure, and average accuracy compared to other approaches.

The suggested approach greatly enhanced our model’s performance by teaching it the minute features required for precise tiny fire detection. We conducted a thorough evaluation of our revised model’s effectiveness through extensive experimentation and validation. Using a variety of datasets, including pictures with small-fire incidents and difficult scenarios like nighttime settings with hazy lamps or fire-like bulbs, we carried out extensive testing. The experimental results verified a significant improvement in our model’s functionality. For instance, our approach produced recall and precision values of 99.4% and 99.3%, respectively, which are substantially better than those of earlier models—including our own earlier CNN model. The successful application of contour analysis and color characteristics is indicated by the appreciable improvement in accuracy and precision metrics. These outcomes, especially in difficult situations, show how reliable and robust our methodology is. For real-world fire detection applications, it is essential that it can discriminate minor fires from regions that resemble fires and handle complicated situations with accuracy.

Our study’s conclusions provide several directions for further investigation. To increase the model’s performance in even more difficult situations, including really bad weather or highly obscured settings, other improvements should be investigated.

To build a complete fire warning and monitoring system, research can also concentrate on integrating the model with additional sensors and data sources, such as temperature sensors and smoke detectors.

## 5. Limitations

Even though our fire detection model has shown encouraging results, there are still several issues that need to be addressed to fully comprehend both its strengths and its shortcomings.

The main method of our existing model is the detection of visible flames in images. In real-world situations, smoke frequently appears before flames are seen. In these situations, the first smoke may mask the flames, making it difficult for the system to reliably identify the fire. This restriction is made worse in enclosed areas where smoke spreads quickly and blocks cameras installed in the ceiling, making fire detection less effective.The model’s capacity to identify fires in circumstances where flames are invisible is restricted by the fact that the dataset utilized in this study only includes images of flames. Due to this limitation, it might be challenging to detect fires in their early stages, when there may only be smoke.Hydrogen gas fires present a significant challenge for our current detection method due to the absence of visible flames. Hydrogen fires emit very little to no visible light, which makes it very difficult for a flame-based detection system to reliably detect such flames. This drawback emphasizes the necessity of supplemental detection techniques to our current approach, such as thermal imaging or gas detection sensors.

## 6. Conclusions

Advances in fire detection based on the use of deep learning models show themselves to be a promising safety technology to reduce fire hazard. The focus of our work was to address the limitations of the existing fire detection algorithms in terms of recognizing fire-like artifacts, and, in the case of small-fire detection under challenging conditions, such as night-time with hazy or fire-like lightbulbs. We have achieved significantly high fire detection accuracy by meticulously selecting, refining, and deploying some of the latest CNN architectures. This conclusion provides a detailed overview of our results, the proposed methodology and implications of the study. Our refined CNN model indicates better performance compared to some of the state-of-the-art algorithms and our previous model. We achieved 99.3% precision, 99.4% recall and 99.5% F1-score using our method. These values are higher compared to the existing popular models such as ResNet, VGG16, Faster R-CNN, AlexNet and Dilated CNNs.

The obtained high precision and recall indicate that our model not only reduces false positives and false negatives but also works efficiently for fire incidents. This is crucial for real-world deployments, as missed detections or false activations may lead to catastrophic effects. The significantly improved ability of our model to detect small flames is one of the major gains of this work. We have achieved this by selectively adding some small-fire cases to our training data and employing advanced approaches such as contour and color characteristic analysis. We provided the algorithm the photos of tiny fires to learn local characteristics, which in turn leads to accurate detection in cases where fires are not observable. This property is crucial for practical deployment as it ensures that the model works reliably in the presence of varying illumination conditions and potentially false activations.

To build on the strengths of our current model, future research will focus on incorporating smoke detection capabilities. By integrating smoke detection, we aim to identify flames at earlier stages, which could facilitate quicker and more accurate responses to fire incidents. Smoke detection will complement our existing flame detection capabilities, creating a more robust system that can handle a broader range of fire scenarios and improve overall detection reliability. Additionally, we plan to adapt and deploy our enhanced model within real IoT devices. This transition will enable the practical implementation of our fire detection system in various environments, including residential, commercial and industrial settings. Optimizing the model for IoT platforms will involve addressing challenges related to computational resource constraints and ensuring efficient operation on edge devices. This approach will facilitate real-time monitoring and alerting, making fire detection systems more accessible and effective in practical applications.

## Figures and Tables

**Figure 1 sensors-24-05184-f001:**
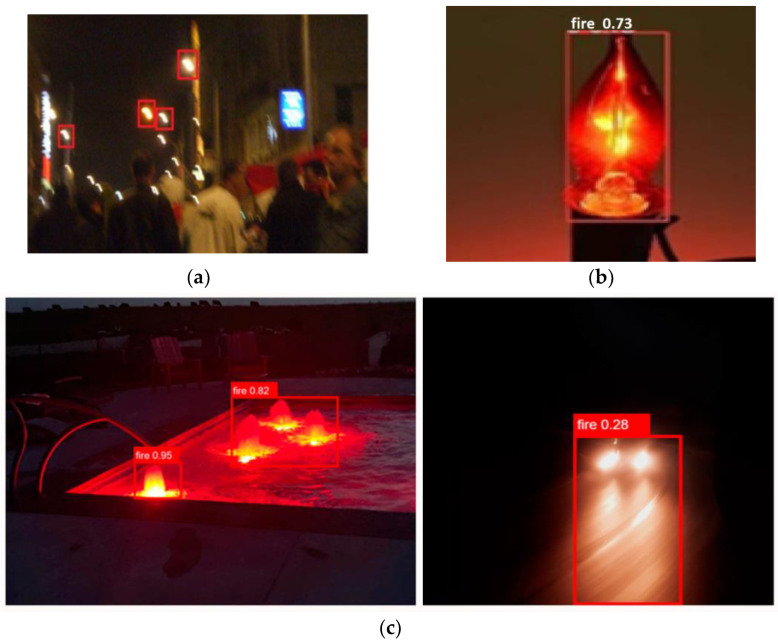
Fire like images: (**a**) Blurred lamps in nighttime environments; (**b**) fire-like bulb as fire; (**c**) non-fire images at night-time.

**Figure 2 sensors-24-05184-f002:**
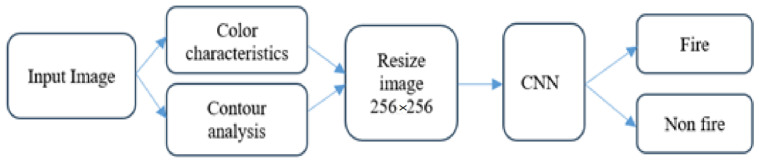
Overall flowchart of technique.

**Figure 3 sensors-24-05184-f003:**
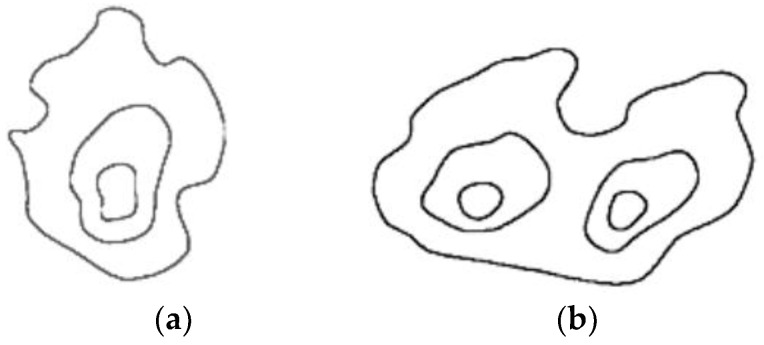
Spatial structure of the flame: (**a**) flame contour with one source; (**b**) flame contour with two centers.

**Figure 4 sensors-24-05184-f004:**
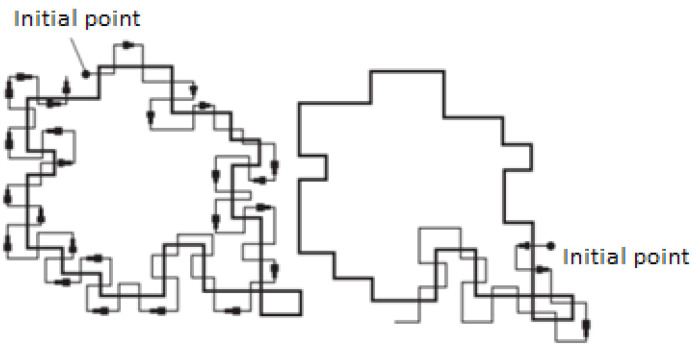
Contour analysis method.

**Figure 5 sensors-24-05184-f005:**
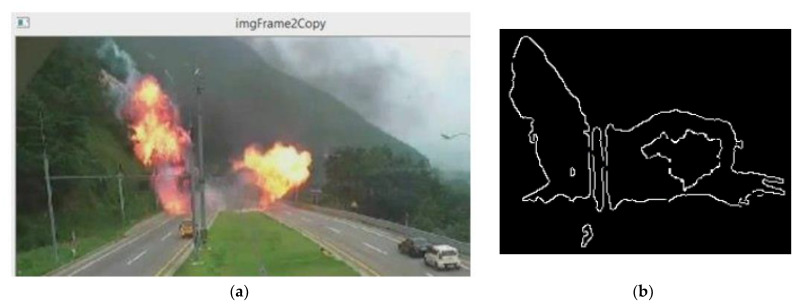
(**a**) Original input image; (**b**) Outline image of fire with two rings.

**Figure 6 sensors-24-05184-f006:**
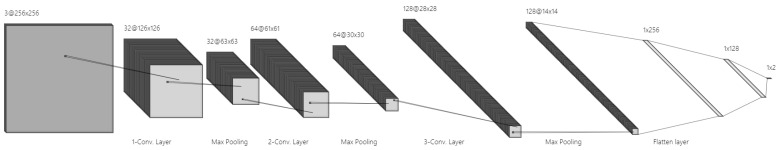
Architecture of CNN Model.

**Figure 7 sensors-24-05184-f007:**
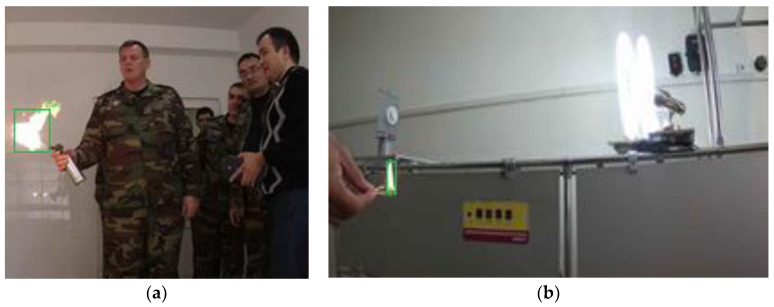
Footage of laboratory experiments: (**a**) recognized fire with a large flame; (**b**) recognized small size fire against the background of a fluorescent lamp.

**Figure 8 sensors-24-05184-f008:**
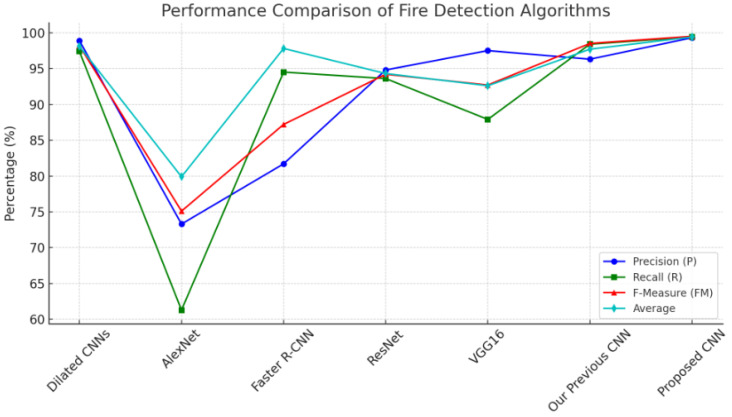
Performance comparison.

**Figure 9 sensors-24-05184-f009:**
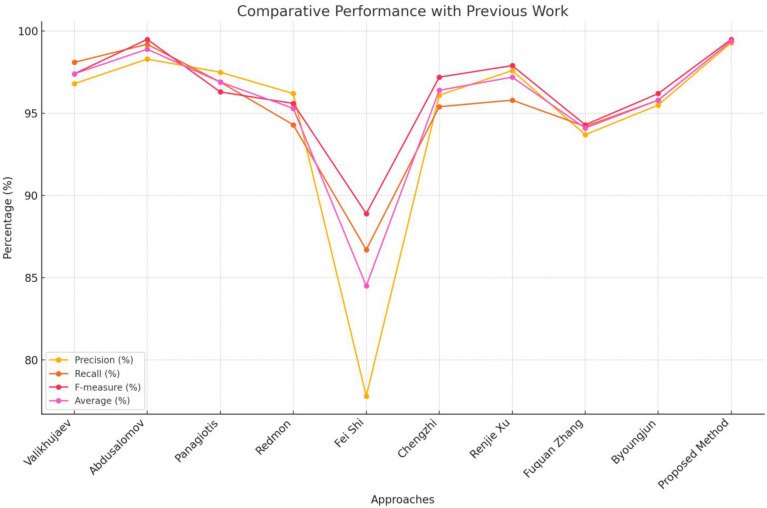
Performance comparison.

**Table 1 sensors-24-05184-t001:** Recognition results for video sequences.

Description	Total Number of Frames	Number of Frames with Flame	Color and Contour Analysis Algorithm			CNN
			FRR	FAR	Result (%)	
Fire in the kitchen	1134	745	24	30	95.2	100
Fire in the supermarket	740	520	5	12	97.8	99.8
Fire in the forest	1560	975	18	22	97.4	99.9
Day lamp and fire	2247	1450	8	2	93.6	99.8
Fire and a man in yellow-red clothes	1487	950	27	48	94.9	99.9
Candle with a dark background	1726	840	35	14	97.2	99.6
Fire on the highway	1578	750	52	84	91.4	99.8
Fire in the mountains	1840	1050	52	56	94.1	99.6
Lamp at night	520	240	10	2	97.7	99.8
Explosion	340	180	4	6	97.1	100
Burning car	680	340	16	18	95.0	100

**Table 2 sensors-24-05184-t002:** Performance Comparison of Fire Detection Algorithms.

Algorithms	P (%)	R (%)	FM (%)	Average (%)
Dilated CNNs	98.9	97.4	98.2	98.1
AlexNet	73.3	61.3	75.1	79.9
Faster R-CNN	81.7	94.5	87.2	97.8
ResNet	94.8	93.6	94.2	94.3
VGG16	97.5	87.9	92.7	92.6
Our Previous CNN	96.3	98.4	98.5	97.7
Proposed CNN	99.3	99.4	99.5	99.4

**Table 3 sensors-24-05184-t003:** Comparative performance with previous work.

Approaches	P (%)	R (%)	FM (%)	Average (%)
Valikhujaev et al. [11]	96.8	98.1	97.4	97.4
Abdusalomov et al. [1]	98.3	99.2	99.5	98.9
Panagiotis et al. [49]	97.5	96.9	96.3	96.9
Redmon et al. [50]	96.2	94.3	95.6	95.3
Fei Shi et al. [51]	77.8	86.7	88.9	84.5
Chengzhi Cao et al. [52]	96.1	95.4	97.2	96.4
Renjie Xu et al. [53]	97.6	95.8	97.9	97.2
Fuquan Zhang et al. [54]	93.7	94.2	94.3	94.1
Byoungjun et al. [33]	95.5	95.8	96.2	95.8
Proposed Method	99.3	99.4	99.5	99.4

## Data Availability

The data presented in this study are available on request from the corresponding author.

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
