# Peer review of "Improving Fire Detection Accuracy through Enhanced Convolutional Neural Networks and Contour Techniques"

_sensors, 2024, doi:10.3390/s24165184_

Round 1
Reviewer 1 Report
Comments and Suggestions for Authors
In this study, a novel method combining contour analysis with a deep CNN is applied for fire detection. The color properties and shape of the fires were utilized. The results of the experiment showed that our improved CNN model outperformed other networks.
1. The introduction should be refined to focus on fire detection. We did not observe a gap in the research on fire detection algorithms.
2. Please give more comment on why Convert the original RGB image to the HSV color space.
3. The authors did not discuss the model parameters proposed in this work.
4. Please indicate the software package used for your program redevelopment.
Author Response
|
Comments 1: The introduction should be refined to focus on fire detection. We did not observe a gap in the research on fire detection algorithms.
|
|
Response 1: Thank you for your valuable feedback. We understand the need for a more focused introduction on fire detection and the identification of research gaps in existing algorithms. We have refined the introduction to emphasize these aspects more clearly. We have updated the manuscript by adding information in the lines from 31 to 51 and in the end of Introduction section. It is highlighted in the lines from 93 to 134.
|
|
Comments 2: Please give more comment on why Convert the original RGB image to the HSV color space. |
|
Response 2: Thank you for your insightful comment. We appreciate the opportunity to clarify the rationale behind converting the original RGB image to the HSV color space. This conversion plays a critical role in enhancing the accuracy and robustness of our fire detection algorithm. We have added detailed explanations to the manuscript to address this point. We updated the manuscript by revising the manuscript. It is highlighted in lines from 457 to 469.
|
|
Comments 3: The authors did not discuss the model parameters proposed in this work.
|
|
Response 2: We agree. Thank you for pointing out the need to discuss the model parameters. We agree that providing detailed information on the model parameters is essential for the clarity and reproducibility of our work. We have revised the manuscript to include the model parameters used in our proposed CNN for fire detection. It is highlighted in lines between 618 and 632.
|
|
Comments 4: Please indicate the software package used for your program redevelopment.
|
|
Response 2: We Agree. Thank you for your feedback. We appreciate the suggestion to indicate the software package used for our program redevelopment. We have added the relevant information to our manuscript. It is highlighted in lines from 405 to 417.
|
Reviewer 2 Report
Comments and Suggestions for Authors
This manuscript explains an improved fire detection method using Convolutional Neural Networks. However, the authors do not appropriately reflect the fire characteristics in their delivery of the fire detection algorithm. Please consider the following points.
1. The literature review in Section 2 is too long; only the key points that are directly relevant to this study need to be explained.
2. There is no explanation of Figure 5 provided in the text.
3. When a fire starts, sometimes the flames come first, but often they come after the initial smoke. In such cases, the flames are expected to be obscured by the smoke, making it difficult to detect the fire. In addition, if the room is enclosed, smoke can spread quickly and obscure ceiling-mounted cameras, thereby making it difficult to effectively detect the fire.
4. Since this study used flame photos as its dataset, it can only detect the flame itself, which makes it challenging to detect the actual fire. Moreover, in the case of hydrogen gas fires, applying this method is expected to be difficult due to the absence of a visible flame.
5. As you mentioned in the conclusion section, missed detections can have adverse effects. In Table 1, you can see that the FRR differs significantly between outdoor areas (highway, mountain) and indoor areas (kitchen, supermarket), with forests having a smaller FRR than other outdoor areas. Why is there such a difference?
6. After initially defining a Convolutional Neural Network (CNN) as CNN, you'll need to use the acronym consistently.
Comments on the Quality of English Language1. Please check the typo-errors such as line 72, 132, 143, 340, 570, and so on.
2. From line 291 to 305, please check your writing.
Author Response
|
Comments 1: The literature review in Section 2 is too long; only the key points that are directly relevant to this study need to be explained.
|
|
Response 1: We agree. We appreciate the reviewer's feedback regarding the length of the literature review in Section 2. We understand the importance of conciseness and relevance in presenting background information. To address this concern, we revised by reducing the related works.
|
|
Comments 2: There is no explanation of Figure 5 provided in the text.
|
|
Response 2: We agree. To address this, we revised the manuscript by adding an explanation. It is highlighted below Figure 5.
|
|
Comments 3: When a fire starts, sometimes the flames come first, but often they come after the initial smoke. In such cases, the flames are expected to be obscured by the smoke, making it difficult to detect the fire. In addition, if the room is enclosed, smoke can spread quickly and obscure ceiling-mounted cameras, thereby making it difficult to effectively detect the fire.
|
|
Response 3: We agree. Thank you for your insightful comment regarding the challenges of detecting fire when smoke obscures flames, particularly in enclosed spaces. In the future work we are planning to solve this issue and the limitations of current fire detection methods. We added limitation section where mentioned this concern. In the conclusion section we mentioned it as our future research plan.
|
|
Comments 4: Since this study used flame photos as its dataset, it can only detect the flame itself, which makes it challenging to detect the actual fire. Moreover, in the case of hydrogen gas fires, applying this method is expected to be difficult due to the absence of a visible flame.
|
|
Response 4: We agree and really did not consider this kind of flame condition in our research. We appreciate your attention to these critical details that influence the effectiveness of fire detection systems. We added a new section discussing the limitations of our current approach and our future research plans to enhance the fire detection system to handle such scenarios effectively.
|
|
Comments 5: As you mentioned in the conclusion section, missed detections can have adverse effects. In Table 1, you can see that the FRR differs significantly between outdoor areas (highway, mountain) and indoor areas (kitchen, supermarket), with forests having a smaller FRR than other outdoor areas. Why is there such a difference?
|
|
Response 5: Thank you for your constructive and attentive feedback. FRR and FAR are errors which were detected during the experiments of performance of Color and Contour analysis separately from CNN. As you see in Table 1 the percentage performance of these methods is not satisfactory. Outdoor environments such as highways and mountains often present challenges due to varying lighting conditions, weather influences, and background complexities. These factors can introduce noise and affect the clarity of fire signatures, leading to higher FRR. In contrast, indoor environments like kitchens and supermarkets typically have more controlled lighting and fewer environmental variables, resulting in lower FRR. The integration of CNN with Color and Contour-based algorithms showed significantly better results as shown in the last column of Table 1.
|
|
Comments 6: After initially defining a Convolutional Neural Network (CNN) as CNN, you'll need to use the acronym consistently.
|
|
Response 5: We agree. We acknowledge the importance of maintaining consistency throughout the manuscript for clarity and precision. We made the necessary revisions to ensure the consistent use of acronyms.
|
|
4. Response to Comments on the Quality of English Language |
|
Point 1: Please check the typo-errors such as line 72, 132, 143, 340, 570, and so on. Point 2: From line 291 to 305, please check your writing.
|
|
Response: We apologize for English grammar mistakes. We have corrected all the mentioned and other grammar errors. They are highlighted in the manuscript.
|

Reviewer 3 Report
Comments and Suggestions for Authors
Dear authors,
I have reviewed the manuscript entitled Improving Fire Detection Accuracy Through Enhanced CNN and Contour Techniques. In this paper, The authors propose a fire detection method based on the combination of contour line analysis and deep learning (CNN). The method mainly uses two algorithms for fire detection: â… : detecting flame color features; â…¡: contour analysis to detect flame shape. In addition, the model is trained with a newly generated dataset in order to be able to better recognize flames or fires. By reading your paper, We learned that the authors' improved model outperforms various currently known models, and the new method improves on all performance metrics with 99.4% accuracy. Also, the authors' proposed method beats many other state-of-the-art methods: Dilated CNNs (98.1% accuracy) Faster R-CNN (97.8% accuracy) and Res-Net (94.3%).Personally, I am interested in this research direction. However, this paper needs to be slightly improved before being accepted for publication.
In general:
1. The results are displayed clearly, but the interpretation of the data can be more in-depth. More in-depth analysis may provide stronger conclusions.
2. The conclusion summarizes the main findings but could include more implications for future research or practical applications.
Details:
1. In Abstract, The accuracy of the new method is 97.7%, which is inconsistent with the 99.4% accuracy of the new method in the later paper; moreover, the 97.7% accuracy of the new method is lower than the 98.1% accuracy of Dilated CNNs. Check for writing errors.
2. What is the role of the contour analysis method here, which does not seem to be covered in the article results, if so, please show it appropriately.
3. For Figures 8 and 9, the authors could have talked about plotting the same parameter for different models as a line graph, so that the advantage of comparison would be more prominent.
Comments on the Quality of English Language
Grammar problem:
1. Page 4 line 142: ‘thru’ needs to be replaced with ‘through’.
2. Page 4 line 143: ‘manage’ needs to be replaced with ‘management’.
3. Page 4 line 159: ‘quick’ needs to be replaced with ‘and quick’.
4. Page 5 line 219: ‘decisor’ needs to be replaced with ‘decision’.
5. Page 5 line 232: ‘Fire’ needs to be replaced with ‘A fire’.
6. Page 7 line 310: ‘CNN’ needs to be replaced with ‘the CNN’.
7. Page 7 line 329: ‘because of preventing’ needs to be replaced with ‘to prevent’.
8. Page 7 line 329: ‘images’ needs to be replaced ‘image’.
9. Page 11 line 484: ‘add’ needs to be replaced with ‘adds’.
10. Page 12 line 502: ‘is provided’ needs to be replaced with ‘provides’.
Author Response
|
Comments 1: The results are displayed clearly, but the interpretation of the data can be more in-depth. More in-depth analysis may provide stronger conclusions.
|
|
Response 1: Thank you for your constructive feedback. We agree that providing a more in-depth analysis of our results will enhance the understanding and significance of our findings. We have added a detailed interpretation of our results, and it is highlighted in section 4.
|
|
Comments 2: The conclusion summarizes the main findings but could include more implications for future research or practical applications.
|
|
Response 2: We agree. To address this, we have expanded the conclusion section to include a comprehensive overview of potential implications for future research and practical implementations of our proposed method. It is highlighted in lines between 901 and 915.
|
|
Comments 3: In Abstract, The accuracy of the new method is 97.7%, which is inconsistent with the 99.4% accuracy of the new method in the later paper; moreover, the 97.7% accuracy of the new method is lower than the 98.1% accuracy of Dilated CNNs. Check for writing errors.
|
|
Response 3: Thank you for pointing this out. We agree with this comment. We have corrected the error.
|
|
Comments 4: What is the role of the contour analysis method here, which does not seem to be covered in the article results, if so, please show it appropriately.
|
|
Response 4: We agree and recognize the importance of clearly demonstrating the contribution of contour analysis to our fire detection methodology. To address this, we have revised the manuscript to explicitly highlight how contour analysis complements the CNN model and enhances overall detection performance. It is highlighted in lines from 683 to 701.
|
|
Comments 5: For Figures 8 and 9, the authors could have talked about plotting the same parameter for different models as a line graph, so that the advantage of comparison would be more prominent.
|
|
Response 5: Thank you for your valuable suggestion regarding the presentation of Figures 8 and 9. We agree that plotting the same parameters for different models as a line graph would enhance the clarity and prominence of the comparison. We have revised the manuscript to incorporate this approach, allowing for a more effective visual representation of the performance differences between models.
|
|
4. Response to Comments on the Quality of English Language |
|
Point 1: Page 4 line 142: ‘thru’ needs to be replaced with ‘through’. Point 2: Page 4 line 143: ‘manage’ needs to be replaced with ‘management’. Point 3: Page 4 line 159: ‘quick’ needs to be replaced with ‘and quick’. Point 4: Page 5 line 219: ‘decisor’ needs to be replaced with ‘decision’. Point 5: Page 5 line 232: ‘Fire’ needs to be replaced with ‘A fire’. Point 6: Page 7 line 310: ‘CNN’ needs to be replaced with ‘the CNN’. Point 7: Page 7 line 329: ‘because of preventing’ needs to be replaced with ‘to prevent’. Point 8: Page 7 line 329: ‘images’ needs to be replaced ‘image’. Point 9: Page 11 line 484: ‘add’ needs to be replaced with ‘adds’. Point 10: Page 12 line 502: ‘is provided’ needs to be replaced with ‘provides’.
|
|
Response: We apologize for English grammar mistakes. We have corrected all the mentioned and other grammar errors. They are highlighted in the manuscript. |
Round 2
Reviewer 1 Report
Comments and Suggestions for Authors
ACCEPT
Author Response
|
Comments 1: Accept
|
|
Response 1: Accept
|
Reviewer 2 Report
Comments and Suggestions for Authors
Please consider the following points.
1. Regarding Response 1, It is reasonable to include the highlights of Section 2 in the introduction, mentioning only those works that are directly relevant to this research.
2. Regarding Responses 3 and 4, considering the importance of fire detection, if the dataset does not reflect real-world fire behavior, selecting an alternative dataset would enhance the value of the CNN model utilized in this study.
3. In response to Response 5, I believe that type 1 errors are more critical than type 2 errors for fire detection. In this manuscript, the variation in FRR observed across different locations is insufficient for effective fire detection purposes.
Author Response
|
Comments 1: Regarding Response 1, It is reasonable to include the highlights of Section 2 in the introduction, mentioning only those works that are directly relevant to this research.
|
|
Response 1: We agree that this will provide better context and clarity regarding the relevance of prior works to our research. We have revised the Introduction accordingly to include a concise summary of the most pertinent related works. The revised Introduction section is highlighted in lines from 43 to 54.
|
|
Comments 2: Regarding Responses 3 and 4, considering the importance of fire detection, if the dataset does not reflect real-world fire behavior, selecting an alternative dataset would enhance the value of the CNN model utilized in this study.
|
|
Response 2: We agree. Thank you for your valuable comment regarding the dataset's reflection of real-world fire behavior. In our current study, the dataset was meticulously curated to include a wide variety of fire scenarios, including both small and large fires, indoor and outdoor environments, and various times of day. However, we acknowledge that real-world conditions can be more complex and varied. To address this concern, in the future we plan to enhance our dataset by incorporating more diverse and representative fire scenarios from publicly available and well-established fire detection datasets. Additionally, we will seek to include data from real-world fire incidents captured by surveillance cameras and other relevant sources. This will ensure that our model is trained and validated on data that more accurately reflects real-world fire behavior. In the conclusion part we mentioned it by highlighting the future work. |
|
Comments 3: In response to Response 5, I believe that type 1 errors are more critical than type 2 errors for fire detection. In this manuscript, the variation in FRR observed across different locations is insufficient for effective fire detection purposes.
|
|
Response 3: We agree. We presented results obtained from contour and color analysis algorithms, which serve as a preliminary step before integrating the CNN. The Type 1 and Type 2 errors are obtained during the experiments Contour and color analyses algorithm. Both results are low. To improve these results we integrated CNN, after that the results are increased. The role of Contour and color analyses algorithm in the integrated model which proposes in this manuscript is detect fire and fire like parts in the input image and select that part by reducing the size of input image. If the Contour and color analyses algorithms cannot fire or fire like parts in the image the input image will be checked by CNN model.
|